# The RHIM of the Immune Adaptor Protein TRIF Forms Hybrid Amyloids with Other Necroptosis-Associated Proteins

**DOI:** 10.3390/molecules27113382

**Published:** 2022-05-24

**Authors:** Max O. D. G. Baker, Nirukshan Shanmugam, Chi L. L. Pham, Sarah R. Ball, Emma Sierecki, Yann Gambin, Megan Steain, Margaret Sunde

**Affiliations:** 1School of Medical Sciences, The University of Sydney, Sydney, NSW 2006, Australia; max.baker@pasteur.fr (M.O.D.G.B.); nirukshan@uow.edu.au (N.S.); chilelan.pham@mq.edu.au (C.L.L.P.); sball@wrentherapeutics.com (S.R.B.); 2EMBL Australia Node in Single Molecule Science, and School of Medical Sciences, University of New South Wales, Sydney, NSW 2052, Australia; e.sierecki@unsw.edu.au (E.S.); y.gambin@unsw.edu.au (Y.G.); 3School of Medical Sciences, Sydney Institute for Infectious Diseases, The University of Sydney, Sydney, NSW 2006, Australia; megan.steain@sydney.edu.au; 4School of Medical Sciences, Sydney Institute for Infectious Diseases, The University of Sydney Nano Institute, The University of Sydney, Sydney, NSW 2006, Australia

**Keywords:** RHIM, TRIF, necroptosis, functional amyloid, fibrils, RIPK

## Abstract

TIR-domain-containing adapter-inducing interferon-β (TRIF) is an innate immune protein that serves as an adaptor for multiple cellular signalling outcomes in the context of infection. TRIF is activated via ligation of Toll-like receptors 3 and 4. One outcome of TRIF-directed signalling is the activation of the programmed cell death pathway necroptosis, which is governed by interactions between proteins that contain a RIP Homotypic Interaction Motif (RHIM). TRIF contains a RHIM sequence and can interact with receptor interacting protein kinases 1 (RIPK1) and 3 (RIPK3) to initiate necroptosis. Here, we demonstrate that the RHIM of TRIF is amyloidogenic and supports the formation of homomeric TRIF-containing fibrils. We show that the core tetrad sequence within the RHIM governs the supramolecular organisation of TRIF amyloid assemblies, although the stable amyloid core of TRIF amyloid fibrils comprises a much larger region than the conserved RHIM only. We provide evidence that RHIMs of TRIF, RIPK1 and RIPK3 interact directly to form heteromeric structures and that these TRIF-containing hetero-assemblies display altered and emergent properties that likely underlie necroptosis signalling in response to Toll-like receptor activation.

## 1. Introduction

TIR-domain-containing adapter-inducing interferon-β (TRIF) is a key innate immune adaptor protein that forms signalling complexes downstream of the activation of Toll-like receptors 3 and 4 (TLR3/4) [1]. TLRs are membrane-associated receptors that are capable of detecting damage-associated and pathogen-associated molecular patterns (DAMPs and PAMPs) [2]. TLR3 detects double-stranded RNA [3], triggering a response to a range of viruses. TLR4 is best known for responding to lipopolysaccharide in gram-negative bacteria [4] and can also respond to gram-positive bacterial and viral PAMPs [5,6,7,8]. As TRIF acts downstream of both TLR3 and TLR4, it plays an important role in inducing immune defence to a wide range of pathogenic stress.

TRIF is a multidomain protein, with each domain capable of signalling for specific cellular outcomes. It contains an N-terminal domain, a TIR (Toll/interleukin1 (IL-1) receptor) domain and a C-terminal Receptor Homotypic Interaction Motif (RHIM) sequence (Figure 1A). The RHIM sequence in the C-terminal region of TRIF has been shown to mediate interactions that can result in the activation of the lytic inflammatory cell death programme known as necroptosis, which is initiated during innate immune stress caused by pathogenic infection [9,10].

Necroptotic death was initially observed downstream of tumour necrosis factor (TNF) receptor ligation, resulting from the interaction of RHIM-containing receptor interacting protein kinase 1 (RIPK1) with the RHIM-containing protein RIPK3 [11,12]. The protein complex formed by the hetero-assembly of RIPK1 and RIPK3 is termed the necrosome [13,14] and is an ultra-high molecular weight functional amyloid signalling complex [15]. The necrosome formed by interactions between RIPK1 and RIPK3 is a heteromeric amyloid assembly driven by interactions between RHIM sequences in these proteins [15].

RHIMs are conserved protein sequences located within disordered regions in four multi-domain immunity-associated proteins (Figure 1A). RHIMs contain a highly conserved core tetrad of residues with the consensus sequence (V/I)Q(V/I/L/C)G (Figure 1B). This core tetrad is important for function, as mutagenesis of this sequence to AAAA in many different experimental workflows ablates RHIM-driven protein function [12,15,16,17,18]. Additionally, the RHIMs are essential for necrosome assembly and amyloid formation: mutation of key amyloid-forming residues in RHIMs results in a loss of necroptosis capability in cells [15]. Recent structural determination of necrosome assemblies has revealed that the RHIM sequence forms the amyloid core of these fibrils, which explains the importance of this sequence for both amyloid assembly and signalling function [19,20].

In addition to RIPK1 and RIPK3, RHIMs that drive cell death have also been identified in the cytoplasmic nucleic acid sensor ZBP1 [17,21,22] and in TRIF. The RHIM from TRIF was first identified as a mediator of apoptosis resulting from TLR activation [23]. Subsequent studies revealed that the RHIM from TRIF could also induce necroptosis by RHIM-driven interaction with RIPK3 in multiple cell types, including macrophages, endothelial cells and fibroblasts [16,24]. The formation of the necrosome and RIPK3 phosphorylation leads to the phosphorylation, conformational change and oligomerisation of the pseudokinase mixed lineage kinase-like protein (MLKL) [25,26,27]. MLKL is the executioner protein of necroptosis [25] and induces cell membrane lysis when activated by RIPK3 [26,27,28].

TRIF is well-established as a pleiotropic adaptor protein involved in Toll-like receptor signalling [1,16], but little is known about the structure of the hetero-assemblies that define its role in programmed cell death. TRIF is capable of signalling for necroptosis in the absence of RIPK1 in murine fibroblasts and endothelial cells; however, it is required for necroptosis in macrophages [16]. The requirement of RIPK1 to modulate TRIF-RIPK3 signalling may be cell-context dependent, and potentially reliant on the presence of other cellular co-factors. The ability of TRIF to form either homo- or heteromeric amyloid assemblies has not been determined, although cell-based experiments have indicated its ability to form insoluble and fibrillar structures in cells [29,30].

Here we have used a range of biophysical approaches to characterise the amyloidogenic capacity of the TRIF RHIM. We have delineated the region of TRIF that forms a protected amyloid-structured core and our results demonstrate that this region extends beyond the 18-residues of the RHIM that are identified by homology. Our results demonstrate the capacity of the TRIF RHIM to form heteromeric assemblies with RIPK1 and RIPK3 through direct interaction.

## 2. Results

### 2.1. The RHIM from TRIF Self-Assembles into Amyloid Structures

A protein construct encoding the RHIM and surrounding residues of wildtype TRIF was cloned with His-tagged ubiquitin or fluorescent partner domains to assist in the expression, purification and detection of amyloid-based assembly and protein:protein interactions (Figure 1C). Protein constructs encoding for RIPK1, RIPK3 and a TRIF core tetrad mutant (VQLG to AAAA) were also cloned (Appendix A). The AAAA mutation was chosen because this change to the core residues of the motif has been shown to reduce the ability of RHIM-containing proteins to signal for cell death by necroptosis.

These protein constructs are named for the partner domain and the stretch of residues incorporated from the original RHIM protein (for instance, Ub-TRIF_601–712_ represents the residues 601–712 of TRIF with an N-terminal ubiquitin partner domain). Proteins were produced by overexpression in *E. coli* and purified by affinity chromatography. Previous experience has shown that RHIM-containing proteins are prone to rapid self-assembly into insoluble aggregates in buffers of physiological salt and pH [31]. Therefore, proteins were purified and maintained as monomers in chaotropic 8 M urea-containing buffer before removal of denaturant, triggering oligomerisation. This methodology has been previously utilised for the characterization of other RHIM proteins [18,20,31,32].

The capability of the TRIF RHIM to drive the formation of homomeric amyloid assemblies was assessed first with a suite of hallmark assays. The fluorescent dye thioflavin T (ThT) and the aniline dye Congo red show characteristic spectral changes when they bind to the cross-β structure in amyloid fibrils and both are widely used to report on the formation of amyloid protein assemblies [33,34]. Both the wildtype and the mutant TRIF-RHIM containing proteins spontaneously form structures that generate enhanced ThT fluorescence emission at 485 nm and induce a red shift in λ_max_ in the absorbance spectrum of Congo red to 540 nm. These features are consistent with the formation of cross-β structure (Figure 2A,B).

The increase in ThT fluorescence in the presence of wildtype TRIF RHIM follows a sigmoidal curve, which is reminiscent of other well-characterised amyloid-forming proteins. The mutant version of the protein displays a modest increase in intensity from a high initial measured point, indicating that rapid assembly into the ThT-binding conformation starts before the first measurement is recorded. There is a difference in the final intensity of the ThT fluorescence between the two proteins but since ThT intensity is not a quantitative measure of amyloid structure and the mode of ThT binding can affect quantum yield, no conclusion can be drawn from these data about the amount of amyloid formed in the two samples.

This experiment was repeated for a range of protein concentrations of wildtype and mutant versions of the TRIF RHIM (Appendix A). For the wildtype, the extent of ThT fluorescence correlated with the concentration of protein and a sigmoidal increase in fluorescence intensity was apparent, consistent with other well-characterised amyloid-forming systems. For the mutant, the ThT fluorescence increased correspondingly with protein concentration, but even the early timepoints showed an intensity above buffer-only, indicative of rapid formation of ThT-binding species.

Ub-TRIF_601–712_ and Ub-TRIF_601–712_mut assemblies both induced a shift in the absorbance maximum exhibited by Congo red, towards 540 nm and similar to that seen with insulin fibrils (Figure 2B), with the mutant form of TRIF yielding a higher absorbance at 540 nm than the wildtype protein. The ThT fluorescence and Congo red data indicate that both wildtype and AAAA TRIF RHIM constructs do form structures with elements of cross-β architecture, albeit with different kinetics of assembly and possibly with a slightly different environment for ThT binding, reflected by the different final ThT intensity.

Static light scattering was performed in parallel with the ThT assays and showed a large increase in scattering signal for Ub-TRIF_601–712_ but little signal for Ub-TRIF_601–712_mut over the course of 60 min at 2.5 µM (Figure 2C). We also examined the concentration-dependence of light scattering for both protein constructs (Appendix A). For Ub-TRIF_601–712_, light scattering increased with concentration, typical of amyloid fibrils which are large insoluble structures. When studying Ub-TRIF_601__–__712_mut, no scattering was visible for the 60 min time period at concentrations between 0.625 µM and 2.5 µM, and was severely attenuated at 5 µM compared to Ub-TRIF_601–712_ at the same concentration. These data demonstrate that the mutant form of the TRIF RHIM does not form large structures and suggest that the VQLG core tetrad is required for the assembly of the TRIF RHIM into large amyloid fibrils.

Transmission electron microscopy (TEM) revealed the difference in the size and morphology of the assemblies formed by the proteins that underly the difference in light scattering signal. Electron micrographs in Figure 2 demonstrate that Ub-TRIF_601–712_ assembles into fibrils that show an unusual higher-level assembly. The long Ub-TRIF_601–712_ fibrils appear to radiate from a central dense core, reminiscent of a sea anemone with tendrils (Figure 2D). The dense cores have diameters 0.5–1 μm, and the tendrils range in length from 0.5 to >3 µm, resulting in an overall diameter of >5 μm. The dense core is composed of very closely intertwined fibrils, while the emanating fibrils are individual fibrils that display a clear twist (Figure 2E). In contrast to the long fibrils formed by the wildtype TRIF sequence, the Ub-TRIF_601–712_mut was observed to assemble into small aggregates 0.5–2 μm in diameter (Figure 2F).

The ThT, Congo red and TEM results demonstrate that the wildtype TRIF RHIM drives the formation of amyloid fibrils which have a propensity to form large clusters. The AAAA mutation abolishes the ability of the TRIF RHIM to form long amyloid fibrils, as evidenced by TEM and light scattering. The Ub-TRIF_601–712_mut forms only small irregular aggregates. Some residual elements of cross-β structure may remain in these aggregates, as evidenced by the concentration-dependent increase in ThT signal at 485 nm and increase in absorbance at 540 nm in the Congo red absorbance spectrum. However, the data show that the intact VQLG core tetrad of TRIF is required for this RHIM to support the formation of long, regular amyloid fibrils, suggesting that amyloid fibril formation is important for the function of the RHIM within the necroptosis signalling pathway.

### 2.2. Identifying the Protected Core of TRIF Amyloid Assemblies

Recent structural elucidation of functional amyloids involved in cellular signalling processes, including the RIPK1-RIPK3 necrosome and Orb2 assemblies that are associated with memory formation in *Drosophila*, has revealed that these fibrils are composed of an amyloid structured core scaffold and peripheral partner domains that remain active in solution [19,35]. We wished to identify the region of TRIF that forms the core amyloid scaffold. Other investigations of amyloid-forming proteins have used protease digestion experiments to identify the stable, hydrogen-bonded scaffold [36]. The stable secondary structure and tight interdigitation of side chains in this scaffold gives relative protection against proteolysis compared to distal flexible regions.

Here we have exposed TRIF-containing amyloid fibrils to digestion by subtilisin, a non-specific peptidase, and then have used mass spectrometry to identify the residual sequences, which represent the amyloid core. ThT-positive fibrils formed by YPet-TRIF_601–712_ were utilised for these experiments, as the partner protein was resistant to subtilisin degradation and hence did not contribute to the insoluble, proteolysis-resistant fraction. Fibrils formed by YPet-TRIF_601–712_ were treated with subtilisin, which led to the rapid release of the YPet domain into the soluble fraction, while the insoluble fraction contained multiple protease resistant fragments, with the most dominant ~6 kDa (Figure 3A). This fragment was subjected to formic acid treatment to achieve depolymerisation (Figure 3B) and then analysed by LC-MS.

The stable amyloid-forming region of TRIF was successfully identified in this way (Figure 3C). The region of the wildtype TRIF protein used as the starting material in these experiments is 112-residues long. From this region, up to 44 residues flanking the core tetrad were conserved in close to 70% of peptide reads, spanning from S654 to Q709. The core tetrad of VQLG was present in every sequence detected. Regions immediately adjacent to the core tetrad are highly represented, with slight decreases in protection from proteolysis in regions more distal to the tetrad. The full list of peptide sequences successfully identified, including any posttranslational modifications, is provided in Appendix A. This extent of residues potentially involved in amyloid formation by TRIF was surprising, as Waltz and Tango algorithms do not predict any amyloidogenic sequences within this TRIF C-terminal region. AmylPred identifies amyloidogenic propensity in the nine residues on the N-terminal side of the tetrad (PLIIHHAQM). All post-translational modifications corresponded to oxidation of methionine residues, which likely occurred as part of the electrospray ionisation process [37].

These data indicate that the VQLG sequence and its immediately flanking regions form the most protected, and hence structured, component of TRIF amyloid fibrils. Alignment of these protected regions in TRIF with the RHIM region previously identified by other groups, delineated with mutagenesis and functional assays, pulldown interaction studies, measures of heteromeric amyloid formation and solid state NMR [13,15,23], indicates that the amyloid-structured region of TRIF is relatively large and likely extends beyond the ~18-residue sequence that controls interactions between the different RHIM-containing proteins (Figure 3C). While the full extent of this structured region may not be necessary for interactions with other proteins, this first unbiased analysis of the amyloid-forming region of the TRIF protein suggests that a large segment of the C-terminal region of TRIF becomes structured when the protein self-assembles.

### 2.3. TRIF Directly Interacts with RIPK1 and RIPK3 to Form Heteromeric Assemblies

Experiments conducted in cells have indicated that assembly of TRIF into large macromolecular structures is vital for the engagement of necroptosis [16,30]. Confocal coincidence spectroscopy (CCS) has previously been used successfully to probe the formation of large amyloid-based homomeric and heteromeric assemblies [31,38]. In CCS, initially monomeric RHIM-containing fluorescent fusion proteins (Figure 1C; Appendix A) are mixed together and then analysed in a ~1 fL confocal volume, allowing for characterisation and quantification of individual molecules and any complexes formed. Fluorescence emission intensity is recorded from this volume over time. Baseline fluorescence arises from monomeric protein in solution. Deflections from baseline in one or both fluorescence channels indicate the presence of higher-order protein assemblies. Co-incidence of signals from the two channels reflects heteromeric assembly [38].

CCS was performed on YPet-TRIF_601–712_, YPet-TRIF_601–712_mut, mCherry-RIPK1_497–583_ and mCherry-RIPK3_387–518_ alone and in pairs. Fluorescence data were recorded each millisecond for three-minute intervals. A representative 250 ms trace for each experimental condition is shown in Figure 4A.

For YPet-TRIF_601–712_, many large homomeric assemblies were apparent. No such large homomeric assemblies were detected when YPet-TRIF_601–712_mut was incubated under amyloid assembly-permissive conditions. mCherry-RIPK1_497–583,_ and mCherry-RIPK3_387–518_ also formed homomeric assemblies under these conditions (Figure 4A).

When YPet-TRIF_601–712_ and mCherry-RIPK1_497–583_ were incubated together under assembly-permissive conditions, many large peaks were detected in the YPet channel, and multiple smaller peaks were observed in the mCherry channel (Figure 4B). This interaction appears to be dependent on the core tetrad of TRIF, as no large peaks were detected when -YPet-TRIF_601–712_mut and mCherry-RIPK1_497–583_ were co-incubated (Figure 4B). For co-incubations of -YPet-TRIF_601–712_ and mCherry-RIPK3_387–518_, many peaks were detected in both the YPet and mCherry emission channels, reflecting the high propensity of both proteins to assemble (Figure 4B). In the mixture of YPet-TRIF_601–712_mut and mCherry-RIPK3_387–518_, many peaks were detected in the mCherry channel, but few were detected in the YPet channel (Figure 4B).

It is possible to utilise quantitative analysis to measure the interaction capability of YPet and mCherry proteins in this CCS system. One such measure of interaction propensity is termed ‘*colourQ*’ and defined as follows:(1)$$colourQ=\frac{number\;of\;Colour\;A\;peaks\;aligned\;with\;Colour\;B\;peaks}{total\;number\;of\;Colour\;A\;peaks}$$

RedQ measurements (which describe the proportion of green peaks aligned with red peaks) were performed on the two-protein pairs described above (Figure 4C). For co-incubations of mCherry-RIPK1_497–583_ with YPet-TRIF_601–712,_ the peaks were highly co-localised (RedQ = 0.9). For mixtures of YPet-TRIF_601–712_mut and mCherry-RIPK1_497–583_, a RedQ score of 0.11 was reported, indicating a low level of interaction. These data indicate that the TRIF VQLG core tetrad motif is important for co-assembly with RIPK1. For mixtures of YPet-TRIF_601–712_ and mCherry-RIPK3_387–518_, these co-incubated proteins showed a RedQ score of 0.40. The interaction of TRIF with RIPK3 appears dependent on the VQLG core tetrad of TRIF, as coincidence of detection between YPet-TRIF_601–712_mut and mCherry-RIPK3_387–518_ was low (RedQ = 0.13; Figure 4C). The RHIM from TRIF appears more amenable to interaction with RIPK1 than RIPK3 (RedQ = 0.9 vs. RedQ = 0.4). This may be due to the higher propensity of RIPK3 to self-assemble, which may reduce its likelihood for heteromeric interaction. 

The number and size of complexes detected in each sample was also analysed by binning the detected fluorescent signals according to intensity and plotting the number of peaks of each intensity range, in a photon counting histogram (PCH; Figure 4D). Comparison of the PCHs for YPet-TRIF_601–712_ and YPet-TRIF_601–712_mut with mCherry-RIPK1_497–583_ shows that the average size of RIPK1-containing complexes increases in the presence of YPet-TRIF_601–712_ (black) compared to RIPK1 alone (magenta), indicating that hetero-assembly with TRIF drives RIPK1 into larger complexes. This change in peak size distribution is not detected when mCherry-RIPK1_497–518_ is co-incubated with the mutant form of TRIF (grey). This confirms the crucial role of the core tetrad in RHIM-based higher order assembly. Analysis of mixtures of TRIFwt and RIPK3 shows that on average these heteromeric complexes (black) are slightly larger than those formed by the RIPK3 RHIM construct on its own (magenta). Additionally, a small number of larger heteromeric complexes (1500–2000 photons/ms; Figure 4D right panel) are observed. This is consistent with a reduced interaction propensity and lower RedQ for this complex compared to the TRIF:RIPK1 complex. No change in average size of oligomer was detected when the mutant form of the TRIF RHIM was co-incubated with mCherry-RIPK3_387–518_, confirming the dependence on an intact wildtype tetrad sequence for interaction between the RHIMs of TRIF and RIPK3. Taken together, these data indicate that the RHIM of TRIF supports complex formation with RIPK1 and RIPK3 and co-assembly with TRIF recruits RIPK1 into macromolecular complexes that are larger than homomeric RIPK1 assemblies.

### 2.4. Hetero-Assemblies Formed between TRIF and RIPK1 or RIPK3 Are Larger and More Stable Than Homo-Assemblies Formed by These Proteins

Sodium dodecyl sulphate agarose gel electrophoresis (SDS-AGE) experiments were used to probe the native size and stability of TRIF homomeric and heteromeric assemblies containing TRIF and one of either RIPK1 or RIPK3. Previous work has shown that the differential stability of RHIM homo- and heteromeric assemblies can be revealed by treatment with 2% SDS [18,31,32].

YPet-TRIF_601–712_ when maintained in 8 M urea remains monomeric (Figure 5A, left panel, lane 1). TRIF assemblies formed following removal of denaturant were retained in the well (Figure 5A, left panel, lane 2). These were confirmed by TEM to be >5 μM in length. These homomeric amyloids formed by TRIF are susceptible to SDS depolymerisation and are depolymerised to oligomers of a range of sizes and monomer (Figure 5A, left panel, lane 3).

mCherry-RIPK1_497–583_ monomer completely traversed to the gel front (Figure 5A, middle panel, lane 1). After dialysis, mCherry-RIPK1_497–583_ migrated partway down the gel, indicating the formation of oligomeric species or small fibrils, which is consistent with the small fibril assemblies made by this protein that we have previously observed by TEM in prior studies (Figure 5A, middle panel, lane 2) [31]. SDS treatment caused the reversion of dialysed mCherry-RIPK1_497–583_ samples to monomer, indicating low stability of RIPK1-only assemblies (Figure 5A, middle panel, lane 3).

When mCherry-RIPK1_497–583_ and YPet-TRIF_601–712_ were co-dialysed, both proteins were retained in the well, and the oligomer band for mCherry-RIPK1_497–583_ was weaker, indicating that TRIF:RIPK1 interaction results in assembly of structures that are larger than the RIPK1-alone structures (Figure 5, right panel, lane 2). These findings are consistent with the observation of larger RIPK1 assemblies by CCS described earlier (Figure 4C). When this mixture was incubated with 2% SDS, a small amount of protein remains visible in the well, along with a streak near the expected oligomer band size (Figure 5, right panel, lane 3), indicating that the hetero-assembly complex containing of RIPK1 and TRIF is partly resistant to SDS, and is more stable than the homomeric assemblies of these two proteins.

A similar set of experiments was performed with YPet-TRIF_601–712_mut and mCherry-RIPK1_497–583_ (Figure 5B). Unlike the wildtype version of YPet-TRIF_601–712_, the mutant formed oligomers (as previously observed by TEM in Figure 2F) instead of fibrils and these were also not SDS resistant (Figure 5B, first panel, lane 3). mCherry-RIPK1_497–583_ alone formed the same oligomer species as previously described (Figure 5B, middle panel) [31]. Co-dialysis of mCherry-RIPK1_497–583_ and YPet-TRIF_601–712_mut did not see them colocalise (no white overlay) nor was any specific change in translocation or stability characteristics visible for either protein (Figure 5B, third panel, lanes 2 and 3). These findings were consistent with confocal CCS, where wildtype TRIF and RIPK1 RHIMs were shown to interact, and it was observed that the wildtype core tetrad of TRIF was required for TRIF to impart changes on RIPK1 (Figure 4B,C).

The homo- and heteromerization of TRIF and RIPK3 were investigated using the same protocols. YPet-TRIF_601–712_ formed large assemblies that were SDS-soluble (Figure 5C, left panel), as previously described. Homomeric mCherry-RIPK3_387–518_ formed large assemblies that were predominantly retained in the well, again consistent with prior studies (Figure 5C, middle panel) [15,31]. Notably, mCherry-RIPK3_387–518_ homo-assemblies were partially resistant to SDS dissolution, with SDS treatment generating large fibrils that migrated only a short distance from the well (Figure 5C, middle panel, lane 3). Analysis of mixtures of TRIF and RIPK3 revealed novel properties induced by heteromerization (Figure 5C, right panel). When dialysed together, YPet-TRIF_601–712_ and mCherry-RIPK3_387–518_ both remained colocalised within the wells and the RIPK3 in these large hetero-assemblies was completely resistant to SDS treatment. The large fibrils observed in the wells could be a mixture of homomeric RIPK3 fibrils, homomeric TRIF fibrils and heteromeric RIPK3:TRIF fibrils. We observe that the stability of a proportion of the RIPK3 and TRIF material is increased, indicating that TRIF and RIPK3 hetero-interaction serves to stabilise both proteins within amyloid assemblies.

These emergent features of TRIF:RIPK3 hetero-assembly are driven by the wildtype core tetrad of TRIF, as evidenced by the lack of co-localisation when these experiments were repeated with YPet-TRIF_601–712_mut and mCherry-RIPK3_387–518_ (Figure 5D, right panel). Oligomers of mutant TRIF and large RIPK3 fibrils were observed at different locations and the mutant TRIF assemblies were depolymerised by SDS.

### 2.5. Microscopy of Hetero-Assemblies Containing TRIF and Partner Proteins Elucidates Extensive Cointegrated Protein Architectures

The hetero-assemblies formed by TRIF and RIPK1 and RIPK3 were imaged by both TEM and fluorescence confocal microscopy to characterise their size, morphology and integration and localisation of the constituent proteins.

Two prominent morphologies were observed by TEM when Ub-TRIF_601–712_ and mCherry-RIPK1_497–583_ were allowed to co-assemble (Figure 6A): a network of fibrils (Figure 6A, Morphology A) and a dense fibril bundle with some smaller fibrils emanating from the core (Figure 6A, Morphology B). The relationship between Morphology A and Morphology B is unclear—it is possible that Morphology A is on a pathway to Morphology B, or that the different morphologies arise from different clusters of each of the constituent proteins in a single superstructure. Assemblies containing YPet-TRIF_601–712_ and mCherry-RIPK1_497–583_ were assessed by confocal microscopy (Figure 6B) and very large clusters of fibrils were observed and appeared to contain both proteins, with YPet and mCherry colocalised (Pearson correlation value = 0.95). This co-assembly was dependent on the presence of a WT tetrad sequence, since samples prepared with both YPet-TRIF_601–712_mut and mCherry-RIPK1_497–583_ present were observed to contain structures composed almost entirely of RIPK1 (Figure 6C), as indicated by the Pearson correlation score of 0.08. No large structures containing YPet-TRIF_601–712_mut were observed.

Co-assembled structures of Ub-TRIF_601–712_ and YPet-RIPK3_387-518_ imaged by electron microscopy had a single prominent morphological identity (Figure 7A) comprised of dense aggregated fibrillar material. Higher-magnification of these TRIF-RIPK3 co-assemblies reveals individual fibrils emanating from the surface and visible on the periphery of the dense bundles (Appendix A). Coincubation of YPet-TRIF_601–712_ and mCherry-RIPK3_387–518_ and subsequent observation by confocal microscopy confirmed both proteins were contained within these assemblies (Figure 7B). The observed assemblies had extensive fibrillar morphology and were highly colocalised (Pearson correlation = 0.95). This co-assembly requires a WT tetrad sequence: in mixtures of YPet-TRIF_601–712_mut and mCherry-RIPK3_387–518_ the proteins were poorly colocalised (Pearson correlation = 0.18). The observed assemblies displayed low levels of YPet-TRIF_601–712_mut incorporation and appeared to be mostly composed of mCherry-RIPK3_387–518_ (Figure 7C). In order to definitively ascertain whether RIPK3-TRIF co-assemblies were indeed hetero-amyloids, we imaged these assemblies in the presence of ThT, using widefield fluorescence microscopy (Appendix A). The large assemblies composed of wildtype YPet-TRIF_601–712_ and mCherry-RIPK3_387–518_, and corresponding to those observed by confocal microscopy (Figure 7B), were highly co-localised with signal for ThT confirming the cross-β amyloid architecture of the assemblies. Mixtures of YPet-TRIF_601–712_mut and mCherry-RIPK3_387–518,_ generated smaller assemblies with a weaker YPet signal but strong mCherry signal, corresponding to low levels of the TRIF mut relative to RIPK3 RHIM and in line with the structures observed using confocal imaging (Figure 7C). These structures were highly colocalised with ThT signal. However, RIPK3 homomeric fibrils would also be expected to bind ThT so the extent of heteromeric TRIF_601–712_mut:RIPK3 heterofibril assembly could not be distinguished.

## 3. Discussion

Previous studies reporting on TRIF have indicated its ability to form large insoluble filamentous aggregates in cells and to interact with RIPK1 and RIPK3 [29,30] and the results presented here provide the first structural characterisation of those RHIM:RHIM interactions between TRIF and key necroptosis-associated proteins. We have demonstrated that the RHIM from TRIF is capable of spontaneously forming fibrillar amyloid assemblies, like other RHIM-containing proteins [15,31,32]. The assembly of the TRIF RHIM-containing domain into organised amyloid fibrils is crucially dependant on the core tetrad VQLG residues of the motif. Although the AAAA mutant forms ThT and Congo red positive structures, the loss of the key interacting residues abrogates the domain’s ability to form ordered and stable fibrils, as observed by SDS treatment followed by agarose electrophoresis, coincidence confocal spectroscopy and confocal and electron microscopy.

The mass spectrometry data presented here reveal that the protected amyloid core of TRIF assemblies is larger than the sequence motif defined by homology to other RHIM proteins and by pulldown experiments [16]. The structures of heteromeric RIPK1:RIPK3 and homomeric RIPK3 RHIM cores, determined by solid state NMR, show that the residues involved in amyloid formation by RHIM proteins can vary to some extent [19,20]. Additionally, both homomeric and heteromeric forms of RHIM-containing proteins may be important in the determination of cell fate [20,39,40,41]. RIPK3 homo-assembles are comprised of a single protofilament composed of three β-strands [20]. RIPK3 hetero-assemblies with RIPK1 form a serpentine fold comprised of two protofilaments [19,41]. The number, identity and importance of the specific amino acid residues that comprise the amyloid core in these two different conformers vary. It is possible that the architecture of amyloid formed by RHIM-containing proteins differs depending on homo-assembly or specific hetero-assembly status. This would be consistent with the characterisation of mosaic RIPK1:RIPK3 necrosomes by Chen et al. (2022) where the supramolecular architecture of the necrosome reflects its functional impact in cells [41]. Elucidation of TRIF amyloid homo-assemblies or hetero-assemblies with RIPK1 and/or RIPK3 by solid state NMR or cryo-EM is an important avenue for future study. These structures would allow for comparison with the elucidated RIPK1:RIPK3 structure [19], and reveal differences in assemblies formed by TRIF compared to RIPK1 or RIPK3 that may underlie physiological function.

The suite of experiments used in this study has revealed that the RHIM of TRIF binds directly to both RIPK1 and RIPK3 RHIMs, with distinct effects on the two interacting partners. When TRIF binds to RIPK1, it drives RIPK1 into heteromeric assemblies that are larger and more stable than RIPK1 homomeric assemblies. When TRIF forms two-protein hetero-assemblies with RIPK3, these are larger assemblies than formed by either protein alone and they are more stable than TRIF homomeric assemblies. The experiments reported here describe the two-protein assemblies formed by either human TRIF and RIPK1 or TRIF and RIPK3 and they form part of the extensive suite of protein interactions that drive programmed cell death [42]. However, in-cell experiments indicate that TRIF signalling for necroptosis may also involve hetero-interactions between all three of TRIF, RIPK1 and RIPK3 in mouse cells [16].

In murine fibroblasts and endothelial cells, TRIF and RIPK3 are capable of inducing necroptosis without the activity of RIPK1, whereas in macrophages RIPK1 is required for signalling [16]. The data reported here indicate that human TRIF is capable of direct interactions with both RIPK3 and RIPK1 but do not provide an explanation for the cell-type- or host-dependence of TRIF signalling outcomes. Therefore, it appears likely that other cellular co-factors are involved in induction of necroptosis cascades in these different cell types. A recent report studying co-factor driven modulation of ZBP1-RIPK3 necroptosis described that caspase 6 binds directly to RIPK3 to facilitate its interactions with ZBP1 [43]. It is possible that caspase 6 could also facilitate interaction between RIPK3 and TRIF, or modulate the amyloidogenicity or stoichiometry of amyloid complexes, rendering them more or less amenable to activation of the downstream pseudokinase MLKL. Little information is known about the cellular concentrations of RHIM proteins during microbial infection, their specific cellular localisation, or any important post-translational modifications, including ubiquitination [44,45,46]. These characteristics may also modulate or control the structural properties of immunity-associated structural assemblies beyond the intrinsic amyloidogenicity and interaction-propensity of the proteins themselves.

Overall, this work has demonstrated that TRIF is capable of forming heteromeric amyloid structures with unique biophysical and biochemical characteristics that may underlie their signalling function. The identification of TRIF as another protein that exerts signalling function by oligomeric assembly adds to the growing number of recognised important supramolecular signalling complexes in immunity, known as signalosomes [47]. These higher-order protein assemblies have specific architectural features that drive the activation of downstream effector proteins [47,48]. They include Fas/FADD complexes for caspase 8 activation [49]; MyD88 and partner proteins in the Myddosome [50]; ubiquitination complexes including TRAF 6 y [51], multiple variants of the inflammasome [52,53,54], CARD9/CARD11 filaments [55] and MAVS filaments [56]. This work, utilising truncated RHIM-containing domains, provides the first step towards understanding of the size, stoichiometry and stability of assemblies generated in cells by full-length RHIM-containing proteins during genuine infection. Future work will focus on the characterisation of multi-protein RHIM-based complexes to understand how these assemblies form, what effects co-factors may have and what role they play in signalling.

## 4. Materials and Methods

### 4.1. Production of RHIM Protein Fragments

Synthetic gene sequences encoding RHIM-containing regions of human RIPK1, RIPK3, TRIF and ZBP1 were purchased from Genscript and cloned into modified pET15b vectors to generate genes for His-tagged fusion proteins with YPet, mCherry or ubiquitin as N-terminal fusion partners for the RHIM domains (as illustrated in Figure 1). Successful cloning was confirmed by sequencing at the Australian Genome Research Facility. Recombinant expression was achieved in *E. coli* BL21(DE3) cells, with induction of gene expression by IPTG (0.5–1 mM) and 2–4 h incubation at 37 °C following addition of IPTG. Harvested cell pellets were stored at −20 °C until purification.

### 4.2. Purification of Proteins by Nickel Nitriloacetic Acid Affinity Chromatography

Cell pellets were resuspended in 20 mM Tris, 150 mM NaCl, 2 mM EDTA pH 8.0 at 5 mL/by shaking at 180 rpm at 37 °C for 30 min, with intermittent vortexing. Cells were lysed on ice by sonication and then centrifuged at 16,000× *g* and supernatant removed. The remaining pellet was washed and solubilised in 8 M urea, 20 mM Tris, pH 8.0 with mixing at 4 °C overnight. Remaining insoluble material was removed by centrifugation at 16,000× *g* for 30 min. β-mercaptoethanol was added to a final concentration of 4.8 mM and the solubilised protein was incubated with Ni NTA bead (agarose, supplier) with gentle agitation for 2 h. Resin was washed with 10 column volumes of 8 M urea, 20 mM Tris, 25 mM imidazole, pH 8 containing 4.8 mM β-mercaptoethanol and proteins were eluted with 8 M urea, 20 mM Tris, 300 mM imidazole, pH 8.0 containing 4.8 mM β-mercaptoethanol. Fractions containing the desired protein were pooled, concentrated to approximately 1 mL using Amicon ultra15 MWCO 10,000 centrifugal units and then buffer-exchanged into 8 M urea, 20 mM Tris, pH 8.0. Protein concentration was determined using the Pierce BCA Protein Assay Kit (ThermoFisher-Scientific, Scoresby, VIC, Australia), using a fresh standard curve prepared with bovine serum albumin in relevant buffer for each set of protein concentration measurements. Where protein stocks contained urea, the equivalent urea concentration was included in the samples used to create the standard curve. In all samples, the urea concentration was adjusted to less than 100 mM before protein determination using this kit.

### 4.3. Thioflavin T Formation and Light Scattering Assays

Buffer containing 25 mM NaH_2_PO_4_, 150 mM NaCl, pH 7.4, 40 µM ThT and 0.5 mM DTT was pre-loaded into black Costar 96 well fluorescent plates (#3631), the top sealed with Corning Plastic Sealing Tape (Product #6575) and incubated to a temperature of 37 °C in a POLARstar Omega microplate reader (BMG Labtech). Amyloid assembly was initiated by diluting desired proteins out of urea-containing buffer to required concentrations in the wells (specific concentrations varied between experiments and are indicated in each case), with final urea concentration below 300 mM. Fluorescence intensity to indicate amyloid formation was recorded with excitation filter 440 nm (+/− 10 nm) and emission filter 480 +/− 10 nm. Aggregate size was determined by simultaneous measurement of scattering of light at 350 nm. Assays were performed in triplicate. Data analysis was performed in Microsoft Excel and GraphPad Prism.

### 4.4. Congo Red Assays

RHIM-containing proteins were dialysed overnight against 25 mM NaH_2_PO_4_, 150 mM NaCl, 0.5 mM DTT, pH 7.4 at 0.3 mg/mL to allow assembly into amyloid. Positive control insulin amyloid fibrils were prepared by heating a solution of insulin at 2.5 mg/mL in 20 mM glycine, pH 2.0 with shaking at 700 rpm for 6 h, followed by 10-fold dilution. Negative control monomeric insulin was prepared at 0.25 mg/mL in 20 mM glycine, pH 2.0 at room temperature. Congo red from a filtered stock solution of 100 μM in 90% 25 mM NaH_2_PO_4_, 150 mM NaCl, 10% ethanol was added to a final concentration of 20 µM and the absorbance spectrum collected over 350–850 nm using cuvettes with 1cm pathlength in a Nanodrop™ 2000C Spectrophotometer (Thermo Fisher Scientific, Scoresby, VIC, Australia), relative to buffer only. The absorbance from a protein-only sample was subtracted from the spectrum arising from Congo red in the presence of each protein.

### 4.5. Confocal Microscopy

Proteins of interest were diluted to 2.5 µM into sodium phosphate buffer into black Costar 96-well fluorescent plates (Corning, New York, NY, USA) and incubated at 37 °C. Amyloid formation in one sample was recorded by addition of ThT and when amyloid formation was complete, as judged by plateau phase of ThT fluorescence, samples without ThT were prepared for confocal imaging. A protein sample measuring 30 µL was dispensed onto a microscope slide and a coverslip carefully lowered to exclude air bubbles before sealing and storage in the dark overnight at 4 °C before imaging on a Leica SP5 Confocal Microscope.

### 4.6. Widefield Fluorescence Microscopy

Proteins of interest were diluted to 2.5 µM into assembly-permissive sodium phosphate buffer (25 mM NaH_2_PO_4_, 150 mM NaCl, 0.5 mM DTT, pH 7.4) into black Costar 96-well fluorescent plates (Corning) and incubated with 40 µM ThT at 37 °C. After two hours, 30 µL of protein sample was dispensed onto a microscope slide and a coverslip carefully lowered to exclude air bubbles before sealing and storage in the dark overnight at 4 °C before imaging on a Cytation 3 Imager (Biotek, Winusky, VT, USA). CFP, GFP and Texas Red filter sets were used to detect ThT fluorescence, YPet and mCherry respectively. Image analysis was performed in ImageJ (Fiji).

### 4.7. Transmission Electron Microscopy

Samples of individual proteins or desired combinations of RHIM-containing proteins prepared in buffers containing 8 M urea were dialysed overnight against 25 mM NaH_2_PO_4_, 150 mM NaCl, 0.5 mM DTT, pH 7.4 at 1 µM concentration. Copper transmission electron microscopy grids with formvar film support and carbon coating (ProSciTech, Kirwan, QLD, Australia) were floated on 20 µL of dialysed protein suspensions for 10 min. Excess solution was discarded, and the grid washed with three droplets of MilliQ water before staining with 2% uranyl acetate. Grids were imaged using a Tecnai T12 microscope operating at 120 kV and images captured with a side-mounted CCD camera and RADIUS 2.0 imaging software (EMSIS GmbH, Münster, Germany).

### 4.8. Subtilisin Digestion and Mass Spectrometry Analysis

YPet-TRIF_601−712_ (50 µM) was dialysed out of 8 M urea buffer into sodium phosphate buffer overnight. Subtilisin was added at 200:1 YPet-TRIF:subtilisin molar ratio for 4 h in order to completely digest proteins. Samples were then separated by centrifugation into soluble and insoluble fractions. The insoluble fraction was resuspended at 1× and 10× concentration in 8 M urea to identify potential insoluble bands comprising the amyloid core. These experiments were repeated in a workflow suitable for mass spectrometry, where insoluble pellets were resuspended in 90% formic acid, freeze dried, and then resuspended in sodium phosphate buffer. The samples were then electrophoresed by SDS-PAGE and bands of interest were excised. The excised samples were trypsin digested and underwent liquid chromatography mass spectrometry to identify peptide sequence.

### 4.9. Confocal Coincidence Spectroscopy

All of the confocal coincidence spectroscopy data were collected on a home built NanoBright Zeiss Axio Observer microscope system, as described in [57]. Proteins, either alone or in pre-mixed combinations, were diluted 100 fold out of 8 M urea, 100 mM NaH_2_PO_4_, 20 mM Tris, pH 8.0 into 100 mM NaH_2_PO_4_, 20 mM Tris, 0.5 mM DTT, pH 8.0, resulting in final protein concentration of 0.175 µM in 80 µM urea. Two lasers (emitting at 488 nm and 561 nm) were focused within the solution and fluorescence emission collected with 525/20 nm bandpass filter and 580 nm longpass filter separated by 565 nm dichroic, for YPet and mCherry signals respectively. Signals were detected simultaneously for both channels and binned at 1 ms intervals. Data analysis was performed using custom routines in MatLab, adapted from workflows described in [38,57].

### 4.10. Sodium Dodecyl Sulfate Agarose Gel Electrophoresis

Proteins of interest were diluted to 5 µM in 8 M urea-containing buffer, either alone or in mixtures. A sample of each was retained for later use as a monomeric, unassembled control. The remaining sample was dialysed against 100 mM NaH_2_PO_4_, 20 mM Tris, 0.5 mM DTT, pH 8.0 overnight. The next day, glycerol was added to samples to a final concentration of 4%, and bromophenol blue to a final concentration of 0.0008%. SDS (from 20% stock solution) or equivalent volume of water, was added to some samples to 2% and then samples incubated for 10 min before electrophoresis through a 1% agarose gel in 40 mM Tris, 20 mM acetic acid, 1 mM EDTA, 0.1% SDS, pH 8.3, at 50 V for 120 min. Gels were imaged on a ChemiDoc (Bio Rad Laboratories, Hercules, CA, USA), where YPet signal was detected using 605/50 nm emission filters, and mCherry signal was detected using 695/55 nm filters.

## Figures and Tables

**Figure 1 molecules-27-03382-f001:**
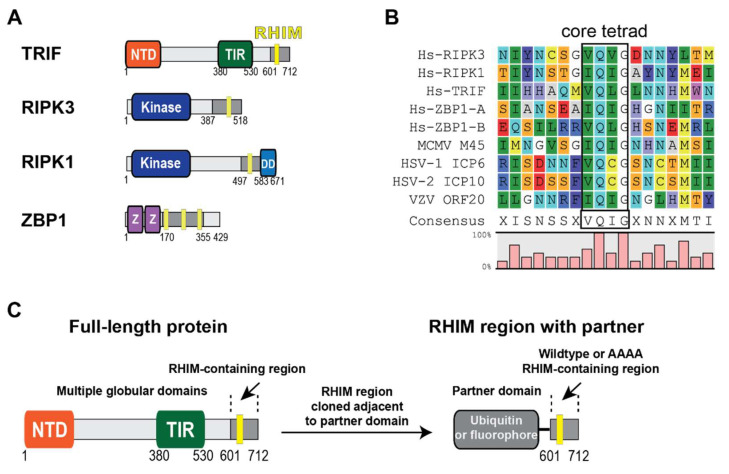
(**A**) Domain architecture of human RHIM-containing proteins, which contain multiple globular domains that serve diverse functional roles. RHIMs are present within disordered protein regions. Z: Zα domain; DD: death domain; NTD: N-terminal domain; TIR: Toll/interleukin-1 (IL-1) receptor domain. (**B**) Alignment of RHIM sequences from human and virus proteins. ZBP1 contains at least two RHIMs, presented as RHIM-A and RHIM-B. The conserved core tetrad is indicated in the boxed region. (**C**) Design of recombinant RHIM-containing fusion proteins with RHIM-containing regions required for necroptosis (light grey) and ubiquitin or fluorescent partner domain.

**Figure 2 molecules-27-03382-f002:**
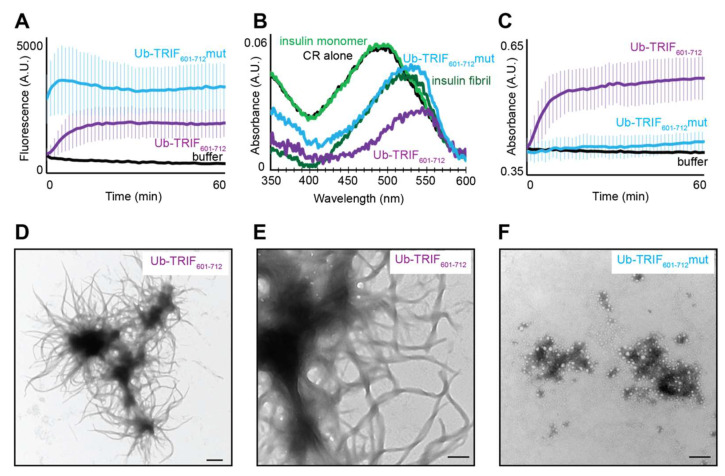
(**A**) Time course of amyloid assembly by Ub-TRIF_601–712_ wild type and mut, initiated by dilution from denaturant into assembly-permissive buffer at 2.5 µM, and assessed by ThT fluorescence. Curves indicate average of three independent replicates, and error bars indicate one standard deviation. (**B**) Congo red absorbance spectra from Ub-TRIF_601__–__712_ WT and mut macromolecular assemblies, compared to insulin monomer, insulin amyloid fibril and Congo red and buffer samples. (**C**) Time course of static light scattering from Ub-TRIF_601__–__712_ RHIM assemblies, following dilution from denaturant into assembly-permissive buffer at 2.5 µM. Curves indicate average from three independent replicates, and error bars indicate one standard deviation. (**D**) Representative transmission electron micrograph of 1 µM Ub-TRIF_601–712_ assemblies depicting ‘sea anemone’ morphology. Scale bar 500 nm (**E**) Representative transmission electron micrograph of 1 µM Ub-TRIF_601–712_ at higher magnification. Scale bar 200 nm. (**F**) Representative transmission electron micrograph of amorphous 1 µM Ub-TRIF_601–712_mut aggregates. Scale bar 500 nm.

**Figure 3 molecules-27-03382-f003:**
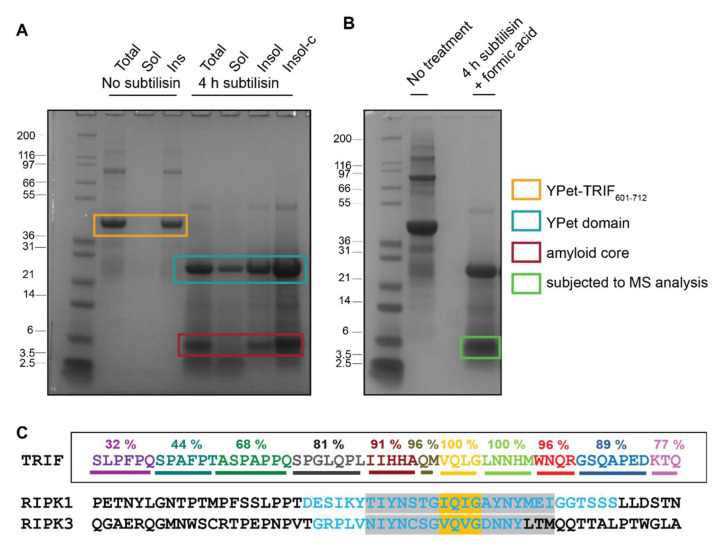
(**A**) SDS PAGE analysis of products of subtilisin digestion of YPet-TRIF_601–712_. Samples are total (Tot), soluble (Sol), insoluble (Insol) or 10X concentrated insoluble (Insol-c). (**B**) Digestion products of YPet-TRIF_601–712_ after lyophilisation, resuspension in 90% formic acid and additional lyophilisation, indicating band excised from gel and subjected to mass spectrometry. (**C**) The sequences of the RHIM-containing regions of human TRIF, RIPK1 and RIPK3 aligned by the core tetrad, coloured yellow. % Prevalence of detected TRIF peptides indicated above sequence. All sequences identified by mass spectrometry available in Appendix A. Regions within RIPK1 and RIPK3 previously identified as important for amyloid interactions [15] and functional RHIM-dependent signalling [13] indicated in blue and grey, respectively.

**Figure 4 molecules-27-03382-f004:**
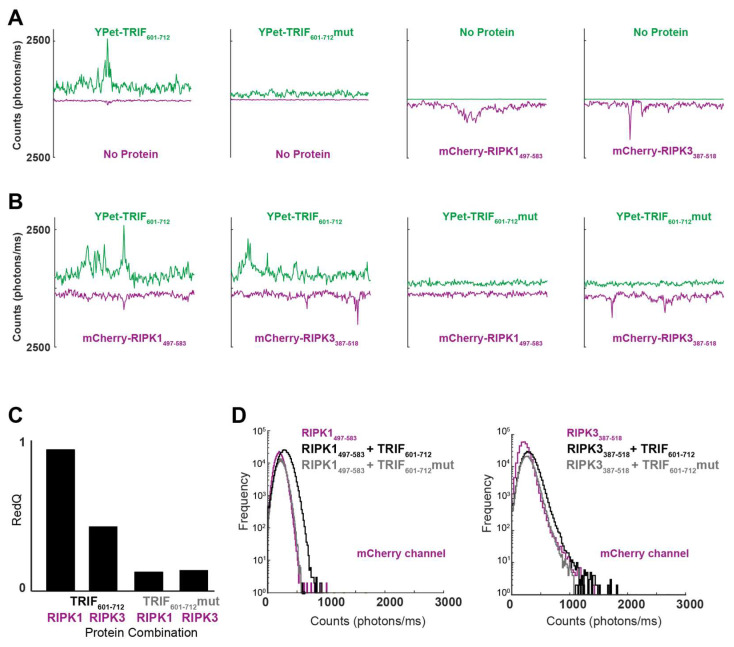
(**A**) Representative 250 ms time traces from individual RHIM-containing proteins at 0.175 µM under assembly-permissive conditions. YPet and mCherry fluorescence emission was measured simultaneously, allowing for detection of coincidence of different RHIM-containing proteins in individual assemblies diffusing through the confocal volume. Proteins in each sample are indicated on each trace. (**B**) Time traces from mixtures of RHIM-containing proteins: TRIF-RHIM_601–712_wt with RIPK1_497–583_ and RIPK3_387–518_ RHIMs, and TRIF-RHIM_601–712_mut with RIPK1_497–583_ and RIPK3_387–518_ RHIMs. (**C**) Comparison of RedQ scores of TRIF-RHIM_601–712_wt, TRIF-RHIM_601–712_mut, RIPK1_497–583_ and RIPK3_387–518_ RHIM combination pairs. (**D**) Photon count histograms from protein assemblies.

**Figure 5 molecules-27-03382-f005:**
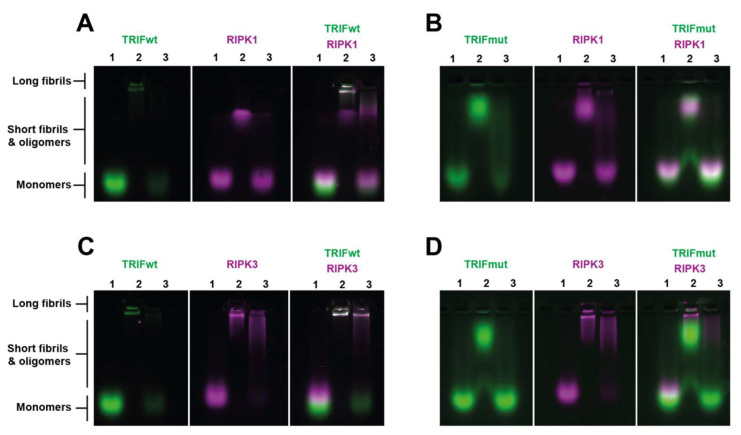
SDS-agarose gel electrophoresis reveals structural properties arising from heteromeric assembly of TRIF and other RHIM-containing proteins. Single or equimolar at 5 µM two-protein combinations carrying YPet or mCherry fluorophores were prepared in (1) monomeric or (2) co-assembled forms and (3) treated with SDS, then electrophoresed on an agarose gel and imaged. Protein combinations used: (**A**) YPet-TRIF_601–712_ and mCherry-RIPK1_497–583_, (**B**) YPet-TRIFmut_601–712_ and mCherry-RIPK1_497–583_, (**C**) YPet-TRIF_601–712_ and mCherry-RIPK3_387–518,_ (**D**) YPet-TRIFmut_601–712_ and mCherry-RIPK3_387–518_. All conditions were imaged with both GFP and mCherry appropriate filter sets, and then overlaid using FIJI ImageJ. All experiments were conducted multiple times, and a representative gel is shown. Migration of long fibrils, short fibrils/oligomers and monomeric forms are indicated.

**Figure 6 molecules-27-03382-f006:**
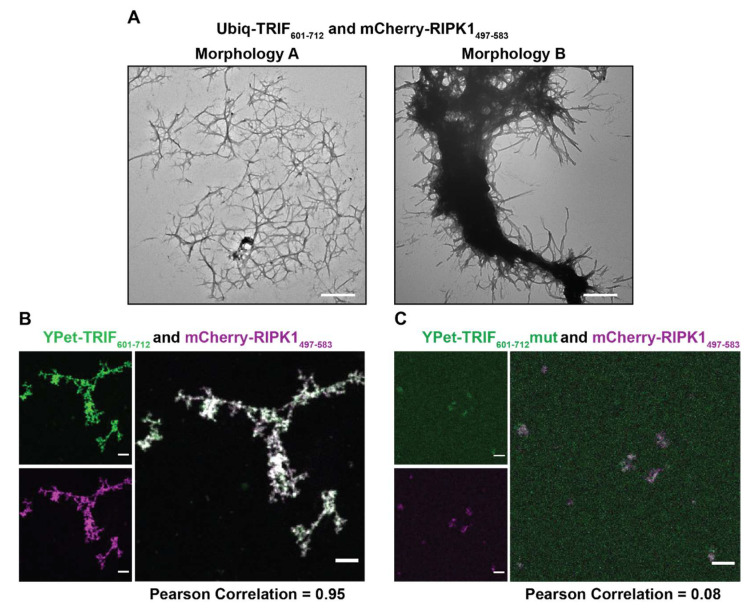
Confocal and transmission electron microscopy of mixtures of TRIF and RIPK1. (**A**) Transmission electron microscopy of mixtures of 1 µM Ub-TRIF_601–712_ and mCherry-RIPK1_497–583_. There were two common morphologies for these mixtures, Morphology A and Morphology B. Scale bars represent 500 nm. (**B**,**C**) Confocal images of mixtures of 2.5 µM TRIFwt_601–712_ and TRIFmut_601–712_ and RIPK1_497–583_ RHIM-containing proteins. Images are displayed as recorded in GFP channel, mCherry channel and merged. Scale bar indicates 10 µm. Image analysis performed in Fiji. Pearson Correlation determined by JaCoP plugin.

**Figure 7 molecules-27-03382-f007:**
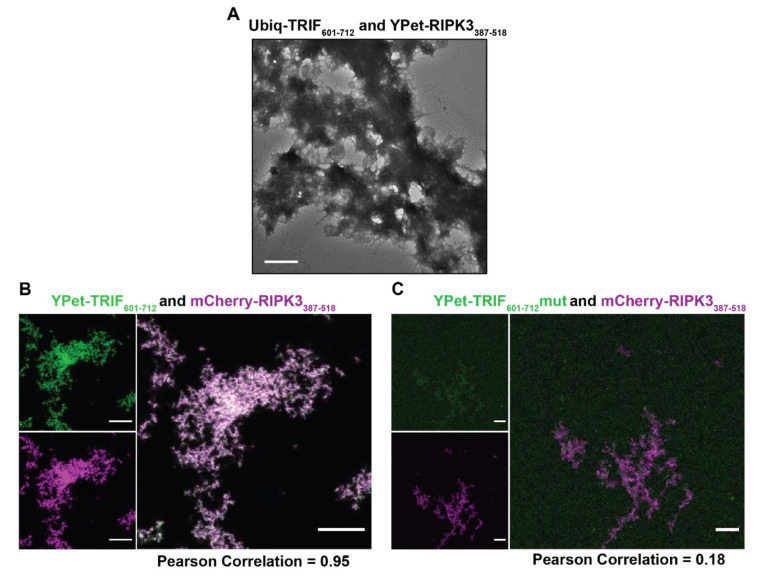
Confocal and transmission electron microscopy of mixtures of TRIF and RIPK3. (**A**) Transmission electron microscopy of mixtures of 1 µM Ub-TRIF_601–712_ and mCherry-RIPK3_387–518_. Scale bars represent 500 nm. (**B**,**C**) Confocal images of mixtures of 2.5 µM TRIF_601–712_wt and TRIF_601–712_mut and RIPK3_387–518_ RHIM-containing proteins. Images are displayed as recorded in GFP channel, mCherry channel and merged. Scale bar indicates 10 µm. Image analysis performed in Fiji. Pearson correlation determined by JaCoP plugin.

## Data Availability

Not applicable.

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
