# Peer review of "The RHIM of the Immune Adaptor Protein TRIF Forms Hybrid Amyloids with Other Necroptosis-Associated Proteins"

_molecules, 2022, doi:10.3390/molecules27113382_

Round 1
Reviewer 1 Report
The manuscript by Max O.D.G. Baker and co-authors provides an insight into damage/pathogen-associated programmed cell death signal transduction pathway and extends the role of functional amyloids in this pathway. The goal of this manuscript is the discovery of amyloidogenic potential of RHIM region of TRIF – one of the components of this transduction pathway. It has been also shown that TRIF can directly interact through RHIM region with downstream components of necroptosis signal transduction pathway RIPK1 and RIPK3 and the region critical for amyloid TRIF aggregation is critical for RIPK1/TRIF and RIPK3/TRIF interaction.
Overall, the article is well written and easy-to-read though some corrections required:
1. The authors postulate that both Ub-TRIF wild type and Ub-TRIF mutant protein are prone to amyloid aggregation (see lines 129-130, 135-136 and 139-141). This conclusion is true for wild type protein but it does not look like significantly substantiated for mutant protein for the following reasons:
- The morphology of mutant Ub-TRIF assemblies does not look like amyloid fibrils (Figure 2F) whereas wild type Ub-TRIF protein fibrils look like amyloidogenic (Figures 2D, 2E).
- Despite the authors talk about small difference in final intensity of the ThT fluorescence between wild type and mutant Ub-TRIF protein (lines 134-135), it does not look like to be true. The increase of fluorescence calculated as final fluorescence intensity at 60 min minus initial fluorescence intensity at 0 min for Ub-TRIF wild type is 2-times higher relative to that for Ub-TRIF mut (Figure 2A).
- Time course of static light scattering for Ub-TRIF mut did not differ from the buffer, whereas in case of wild type protein it differed significantly (Figure 2C) and followed the dynamics of ThT fluorescence (Figure 2A).
- It is not clear if CR absorbance maximum shifts toward to 540 nm for both CR-proteins complexes (Figure2B).
2. There are 5 curves on the Figure2B, but only 3 curves are marked. Marks for both Ub-TRIF proteins are missing.
3. Please, provide an additional proof that Ub-TRIF mut assemblies are amyloidogenic by one of the following techniques:
- Birefringence of polarized light by the complexes of in vitro-made protein assemblies with Congo Red. This is a characteristic feature of amyloid fibrils and, in this case, the easiest way to proof amyloidogenity.
- Immunoreactivity of protein assemblies with anti-amyloid fibril OC antibodies (by dot blot, for example).
4. TRIF/RIPK1 assemblies (Figure 6A) form fibrous structures, whereas TRIF/RIPK3 assemblies looks like clumps, but short fibrils on side of clumps as well (Figure 7A). Could you provide more detailed pictures of the region with short fibrils where fibrous structure of assemblies is recognizable?
Author Response
Reviewer 1:
- The authors postulate that both Ub-TRIF wild type and Ub-TRIF mutant protein are prone to amyloid aggregation (see lines 129-130, 135-136 and 139-141). This conclusion is true for wild type protein but it does not look like significantly substantiated for mutant protein for the following reasons:
The morphology of mutant Ub-TRIF assemblies does not look like amyloid fibrils (Figure 2F) whereas wild type Ub-TRIF protein fibrils look like amyloidogenic (Figures 2D, 2E).
We believe that the ThT and Congo red data collected from wild type and mutant RHIM TRIF proteins do support the conclusion that both form assemblies with a cross-‑b nature at the molecular level. However, while the wild type RHIM assembles into ordered, long amyloid fibrils with characteristic morphology that give rise to light scattering, the AAAA mutant only forms small aggregates which do not generate a large scattering signal. The assembly of the AAAA mutant protein into ThT-positive material occurs even more rapidly than the wild type, possibly reflecting a less ordered ensemble, which cannot undergo higher-order assembly into long fibrils.
We have added data to demonstrate the concentration-dependence of this assembly for both wild type and mutant protein, assessed by ThT fluorescence and light scattering. These data support the conclusion that the AAAA mutant protein can form cross-beta structures in a concentration-dependent manner, but the static light scattering data show that it forms smaller assemblies than the wildtype. We have rewritten this section of the text to clarify these results and the interpretation of them.
Despite the authors talk about small difference in final intensity of the ThT fluorescence between wild type and mutant Ub-TRIF protein (lines 134-135), it does not look like to be true. The increase of fluorescence calculated as final fluorescence intensity at 60 min minus initial fluorescence intensity at 0 min for Ub-TRIF wild type is 2-times higher relative to that for Ub-TRIF mut (Figure 2A).
We thank the reviewer for this observation and have rewritten the text to indicate that the ThT increase over the measured time course is moderate for the mutant. However, we believe that the mutant form of the protein assembles very quickly into small aggregates with some cross-beta architecture formed in the dead time of the experiment. This is consistent with the Congo red results, which show the same shift to lmax at 540 nm for the mutant as for the wild type protein. However, the elements of cross-beta structure in the mutant are not sufficiently ordered and regular to support further higher-order assembly into long regular fibrils.
Time course of static light scattering for Ub-TRIF mut did not differ from the buffer, whereas in case of wild type protein it differed significantly (Figure 2C) and followed the dynamics of ThT fluorescence (Figure 2A).
We agree with this observation. We have now addressed these issues by including additional data and rewriting the section to explain the findings more clearly, indicating that while the wild type protein displays ordered assembly similar to other well-characterised amyloid-forming proteins, the mutant shows a rapid increase in ThT assembly but little scattering, indicative of only small cross-beta structures.
It is not clear if CR absorbance maximum shifts toward to 540 nm for both CR-proteins complexes (Figure2B).
The figure has now been clearly labelled to indicate that indeed both proteins show the shift in absorbance to 540 nm that is characteristic of cross-beta structure. To account for these changes, we have substantially adapted the text to include the following:
Ub-TRIF601-712 and Ub-TRIF601-712mut assemblies both induced a shift in the absorbance maximum exhibited by Congo red, towards 540 nm and similar to that seen with insulin fibrils (Figure 2B), with the mutant form of TRIF yielding a higher absorbance at 540 nm than the wild type protein.
- There are 5 curves on the Figure2B, but only 3 curves are marked. Marks for both Ub-TRIF proteins are missing.
We thank the reviewer, the labels for the Ub-TRIF WT and Ub-TRIF mut samples have now been added to panel 2B.
The text has been rewritten to include a more detailed explanation of the results.
- Please, provide an additional proof that Ub-TRIF mut assemblies are amyloidogenic by one of the following techniques:
Birefringence of polarized light by the complexes of in vitro-made protein assemblies with Congo Red. This is a characteristic feature of amyloid fibrils and, in this case, the easiest way to proof amyloidogenity.
Immunoreactivity of protein assemblies with anti-amyloid fibril OC antibodies (by dot blot, for example).
The Congo red solution assay is a widely accepted alternative to the detection of birefringence (Klunk et al. Methods Enzymol. 1999;309:285-305. doi: 10.1016/s0076-6879(99)09021-7). With purified proteins and the inclusion of appropriate fibrils as a positive control (insulin fibrils as used here), we believe that these data provide strong evidence for the presence of elements of underlying cross-beta structure.
As described in our response to 1, we postulate that the AAAA mutation disrupts the higher-level order that supports long fibrils, although some self-assembly capacity is retained with the remainder of the ~18 residues of this RHIM sequence. Indeed, mutation to Alanine residues may not completely abolish amyloid potential. However, we have chosen this mutation since it has been widely used in functional assays and the functional consequence of this mutation to the tetrad sequence is known.
- TRIF/RIPK1 assemblies (Figure 6A) form fibrous structures, whereas TRIF/RIPK3 assemblies looks like clumps, but short fibrils on side of clumps as well (Figure 7A). Could you provide more detailed pictures of the region with short fibrils where fibrous structure of assemblies is recognizable?
We have provided a new figure (Supplementary Figure S4) where fibrils are visible both on the surface and also extending from the dense fibrillar bundles observed to be formed by TRIF-RIPK3 mixtures.
We have included the following text in the manuscript:
Higher-magnification of these TRIF-RIPK3 co-assemblies reveals individual fibrils emanating from the surface and visible on the periphery of the dense bundles (Supplementary Figure 4).
Reviewer 2 Report
In this manuscript the authors study the amyloid and aggregation capacity of the RHIM domain (111 AA length) of the multi-domain protein TRIF. The authors focus on a 4 amino-acid motif in the domain that is supposed to form the core of the functional TRIF amyloid structure. Furthermore the authors interrogate the formation of inter-protein aggregation between the TRIF RHIM domain and the RHIM domain of RIPK1 and RIPK3.
The article introduces the topic and relevant literature in an adequate way and the results are described. The discussion is a bit speculative and should also focus on the shortcomings of the chosen experimental approaches. The methods section lacks details (see minor comments).
The article claims to have established that the RHIM domain of TRIF forms functional amyloids with the RHIM domain of RIPK3. This claim is not well supported by the data.
Major concerns:
Amyloid formation and protein aggregation is a concentration dependent and higher order process. The authors do not in any figure legend provide details on what protein concentrations of timepoints were used in the assays, this makes interpreting the data unnecessarily difficult.
The method section also does not provide any information on what protein concentration was used in figure 2. The TEM images in figure 2 show a combination of amorphous aggregation and fibril looking aggregation. It appears that two different aggregation mechanisms are involved. This is also supported by the difference in ThT signal (2a) and light scattering (2c) showing a difference in amorphous and amyloid aggregation. It is necessary to conduct the aggregation experiments at a range of protein concentrations to determine the difference in aggregation behavior.
I can't stress enough how important it is to accurately report and measure the used protein concentration when wanting to study and interpret protein aggregation and amyloid formation behavior - it is troubling that on line 520 the authors say that the protein concentration was determined using Pierce Protein assay Kit, but do not specify which Kit is used BCA, Lowery etc. The authors need to clarify how the protein concentration is obtained, because in an 8 M urea buffer some caliometric assays will be a far less reliable measure of protein concentration than others.
In the method section for the confocal coincidence spectroscopy method a range of protein concentrations are mentioned but no protein concentration was given in figure 4, therefore it is impossible to know what is presented in figure 4.
The authors claim that the RHIM domain of TRIF, RIPK1, RIPK3 are forming structured amyloids. This is not supported by the evidence. The image and fluorescence data confirms that these proteins form protein aggregates, which is not surprising for proteins that get diluted together into buffer form denatured states. A lot of different proteins will co-aggregate into a range of morphologies when diluted from high urea concentrations together. The authors need to confirm the formation of amyloids by diagnostics assays such as CD-spectroscopy for the cross-beta structure and ThT, Congo red staining.
Minor concerns:
Figure 2a,2c the error bars are difficult to see.
The authors make a 4 Ala mutation variant of the 4 triad residues (VQLG) as a control for amyloid formation ability of the tested protein. In this particular case I find a 4 Ala mutation a poor choice, because the authors know the structure of the related RIPK3 RHIM structure from Ref. 20 and Ala will be a suitable replacement in this case. The AAA mutation does not prevent the formation of ThT and CongoRed reactive oligomeric proteins. A better control would have been a more informed disrupting mutation such as a Proline for example.
The authors purify the RHIM domain proteins attached to ubiquitin or a fluorescence protein (YPet, mcherry) from inclusion bodies. In that case: Why are the authors not purifying the protein domains without a tag to study the aggregation behavior of the domain in a better controlled reductive approach ?
Line 518 - the authors should check the text in the brackets.
Author Response
Reviewer 2
In this manuscript the authors study the amyloid and aggregation capacity of the RHIM domain (111 AA length) of the multi-domain protein TRIF. The authors focus on a 4 amino-acid motif in the domain that is supposed to form the core of the functional TRIF amyloid structure. Furthermore the authors interrogate the formation of inter-protein aggregation between the TRIF RHIM domain and the RHIM domain of RIPK1 and RIPK3.
The article introduces the topic and relevant literature in an adequate way and the results are described. The discussion is a bit speculative and should also focus on the shortcomings of the chosen experimental approaches. The methods section lacks details (see minor comments).
The article claims to have established that the RHIM domain of TRIF forms functional amyloids with the RHIM domain of RIPK3. This claim is not well supported by the data.
Major concerns:
Amyloid formation and protein aggregation is a concentration dependent and higher order process. The authors do not in any figure legend provide details on what protein concentrations of timepoints were used in the assays, this makes interpreting the data unnecessarily difficult.
The method section also does not provide any information on what protein concentration was used in figure 2. The TEM images in figure 2 show a combination of amorphous aggregation and fibril looking aggregation. It appears that two different aggregation mechanisms are involved. This is also supported by the difference in ThT signal (2a) and light scattering (2c) showing a difference in amorphous and amyloid aggregation. It is necessary to conduct the aggregation experiments at a range of protein concentrations to determine the difference in aggregation behavior.
We thank the reviewer for these comments. We have added a figure to describe the concentration-dependence of ThT fluorescence and static light scattering for both the wildtype and mutant forms of the TRIF RHIM (Supplementary Figure S2). We observe that the ThT signal increases in a concentration-dependent manner for both proteins, which is typical of cross-beta containing structures. Assembly size as determined by static light scattering is concentration-dependent as determined by static light scattering for wildtype, but is severely attenuated at all concentrations of the mutant protein. At 5 µM, there is evidence of formation, but it is much lower than that of the wildtype. These results are consistent with the wild type protein forming long fibrils but the mutant protein forming only small structures, albeit with a cross-beta nature. We have included the details of the concentrations used in the figures and/or figure legends, as appropriate.
We have added text to describe these findings as follows:
This experiment was repeated for a range of protein concentrations of wildtype and mutant versions of the TRIF RHIM (Supplementary Figure S2A). For the wildtype, the extent of ThT fluorescence correlated with the concentration of protein and a sigmoidal increase in fluorescence intensity was apparent, consistent with other well-characterised amyloid-forming systems. For the mutant, the ThT fluorescence increased correspondingly with protein concentration, but even the early timepoints showed an intensity above buffer-only, indicative of rapid formation of ThT-binding species.
And:
We also examined the concentration-dependence of light scattering for both protein constructs (Supplementary Figure 2B). For Ub-TRIF601-712, light scattering increased with concentration, typical of amyloid fibrils which are large insoluble structures. When studying Ub-TRIF601-712mut, no scattering was visible for the 60 min time period at concentrations between 0.625 µM and 2.5 µM, and was severely attenuated at 5 µM compared to Ub-TRIF601-712 at the same concentration. These data demonstrate that the mutant form of the TRIF RHIM does not form structures that are as large as the wild type protein and suggest that the VQLG core tetrad is required for the assembly of the TRIF RHIM into large amyloid fibrils.
I can't stress enough how important it is to accurately report and measure the used protein concentration when wanting to study and interpret protein aggregation and amyloid formation behavior - it is troubling that on line 520 the authors say that the protein concentration was determined using Pierce Protein assay Kit, but do not specify which Kit is used BCA, Lowery etc. The authors need to clarify how the protein concentration is obtained, because in an 8 M urea buffer some caliometric assays will be a far less reliable measure of protein concentration than others.
We thank for the reviewer for this important clarification. Protein concentration was determined by PierceTM BCA Protein Assay Kit (Thermo-Fisher product number 23225). Where protein stocks contained 8 M urea, the equivalent urea concentration was included in the samples used to create the standard curve.
We have added text describing this as follows:
Protein concentration was determined using the Pierce BCA Protein Assay Kit (Thermo Scientific 23225), using a fresh standard curve prepared with bovine serum albumin in relevant buffer for each set of protein concentration measurements. Where protein stocks contained 8 M urea, the equivalent urea concentration was included in the samples used to create the standard curve.
We have also included relevant protein concentrations are indicated in figure legends throughout.
In the method section for the confocal coincidence spectroscopy method a range of protein concentrations are mentioned but no protein concentration was given in figure 4, therefore it is impossible to know what is presented in figure 4.
We thank the reviewer for drawing our attention to this inadvertent omission. The confocal spectroscopy experiments were performed at 0.175 µM for each protein.
We have included the protein concentration in the figure legend and have amended the text describing this as follows:
Proteins, either alone or in pre-mixed combinations, were diluted 100-fold out of 8 M urea, 100 mM NaH2PO4, 20 mM Tris, pH 8.0 into 100 mM NaH2PO4, 20 mM Tris, 0.5 mM DTT, pH 8.0, resulting in final protein concentration of 0.175 µM in 80 µM urea.
The authors claim that the RHIM domain of TRIF, RIPK1, RIPK3 are forming structured amyloids. This is not supported by the evidence. The image and fluorescence data confirms that these proteins form protein aggregates, which is not surprising for proteins that get diluted together into buffer form denatured states. A lot of different proteins will co-aggregate into a range of morphologies when diluted from high urea concentrations together. The authors need to confirm the formation of amyloids by diagnostics assays such as CD-spectroscopy for the cross-beta structure and ThT, Congo red staining.
We have performed widefield fluorescence microscopy of YPet-TRIF and mCherry-RIPK3 co-incubated with ThT and have included the images in Supplementary Figure S5. Proteins assembled in this way co-assemble into morphologies corresponding to those observed by confocal microscopy (Figure 7B and C) and co-localise extensively with ThT. This is strong evidence that these assemblies are hetero-amyloids.
Our data demonstrate that the heteromeric assemblies do have a cross-beta structure. Additionally multiple orthogonal methods confirm specific interactions between the different RHIMs. The confocal coincidence data indicates specific and residue-regulated interaction and assembly (i.e. impacted by AAAA mutation). SDS-AGE reveals novel higher-order properties of these assemblies that are induced by these specific interactions; and TEM data and ThT fluorescence colocalization indicate fibrillar morphology and cross-beta structure characteristic of amyloid.
Definitive proof of the hetero-amyloid structure would require determination of the atomic structure of TRIF-RIPK3 fibrils, by ssNMR or cryoEM, the latter being possible if the fibrils contain a perfectly regular pattern of incorporation of the protein partners. That work lies beyond the scope of this manuscript.
Minor concerns:
Figure 2a,2c the error bars are difficult to see.
We have amended the error bars to make them more visible.
The authors make a 4 Ala mutation variant of the 4 triad residues (VQLG) as a control for amyloid formation ability of the tested protein. In this particular case I find a 4 Ala mutation a poor choice, because the authors know the structure of the related RIPK3 RHIM structure from Ref. 20 and Ala will be a suitable replacement in this case. The AAA mutation does not prevent the formation of ThT and CongoRed reactive oligomeric proteins. A better control would have been a more informed disrupting mutation such as a Proline for example.
We appreciate the reviewer’s comments and agree that AAAA will not necessarily disrupt amyloid assembly entirely. The reason for studying this mutation is that it is a widely recognised functional mutant that has been demonstrated to impair signalling for cell death by necroptosis. While our results show that it does not completely prevent the formation of some element of cross-beta structure, our results provide evidence that functional amyloid activity in this important cellular complex relies on the formation of long regular amyloid fibrils. We believe this functional insight supports the investigation of the impact of this mutation on the structure, properties and interactions of the TRIF RHIM.
The authors purify the RHIM domain proteins attached to ubiquitin or a fluorescence protein (YPet, mcherry) from inclusion bodies. In that case: Why are the authors not purifying the protein domains without a tag to study the aggregation behavior of the domain in a better controlled reductive approach ?
The addition of the tags improves the expression of these highly assembly-proline proteins. Additionally, the use of multiple different fluorescent and non-fluorescent tags allows the investigation of heteromeric amyloids containing two or three different proteins. Notably the tags also mimic the presence of the additional functional domains found in TRIF and RIPK3. The subtilisin digestion experiments demonstrate that the fusion protein partners are readily and rapidly cleaved from the RHIM-containing region, indicating that these fusion partners do not take part in the amyloid assembly driven by the RHIM. Therefore we feel that the use of the fusion proteins is appropriate for this system.
Line 518 - the authors should check the text in the brackets.
In the version we submitted there is no bracketed text on Line 518. We would appreciate clarification about this point? We have expanded on the details of the Pierce Bicinchoninic Acid (BCA) Protein Assay kit supplied by Thermo Scientific.
Round 2
Reviewer 1 Report
The authors answered all the questions asked.
The questions 1, 2 and 4 were answered completely and the text, figures and supplementary materials of the draft were revised correspondingly. Nevertheless, I am not agree with author’s answer on question 3 and have an additional questions about Congo Red assay protocol and the results of Congo Red assay represented on Figure 2B.
Answering question 3, the Authors referred to the manuscript by Klunk et al, published in 1999. Nevertheless, I am not agree that in case Ub-TRIF mut CR absorbance shift is enough to proof its amyloidogenic properties for the following reasons:
- According to recently published Amyloid Nomenclature Guidelines, congophilia and birefringence together remain the gold standard for demonstration of amyloid deposits (Amyloid. 2018. 25(4):215-219. DOI: 10.1080/13506129.2018.1549825). The update of the Guideline explained that the color of birefringence may be not only green, other colors are also possible depending on the orientation of the fibrils in situ. (Benson M.D. et al, Amyloid. 2020. 27(4):217-222. DOI:10.1080/13506129.2020.1835263 and the reference therein A.J. Amyloid. 2019. 26:96. DOI: 10.1080/13506129.2019.1597342).
- Taking into account close-to-zero light scattering by Ub-TRIF mut for concentrations up to 2.5 uM(Fig 2C and Fig S4B), non-fibrillar, amorphous morphology of Ub-TRIF mut assemblies (Fig 2F), amyloidogenic nature of this protein is still controversial. Birefringence analysis will be critical to the proof of amyloidogenic nature of Ub-TRIF mut aggregates the authors have postulated. Positive result of birefrigence test will enforce significantly author’s conclusion, whereas negative result may be a reason to revise it.
For these reasons, birefringence analysis of Ub-TRIF mut or both wild type and mutant protein is critical to make a conclusion about amyloidogenic nature of Ub-TRIF mut protein.
There are two questions about Figure 2B (Congo Red assay).
1 The same concentration of CR (20 uM) were used for CR-alone, CR-insulin monomers, CR-insulin fibrils and CR-UbTRIF aggregates. CR absorbance spectra in the presence or in the absence of insulin monomers almost coincide and this is fine. CR-insulin fibrils curve should also coincide CR alone and CR-insulin monomers curves at the region 300-480 nm and after that (480-600 nm) it should be above them reflecting the appearance of new absorbance maximum of CR-fibrils complex (see Figure 1 and Figure 3 in Klunk WE, Pettegrew JW, Abraham DJ J Histochem Cytochem 1989 37:1273 DOI: 10.1177/37.8.2666510 for the reference). The shape of absorbance spectrum curves for CR-insulin fibrils represented on Figure 2b is correct, but the values of absorbance compared with those for CR alone and CR-insulin monomers look like much lower than they should be. The same situation is for CR-UbTRIF complexes where absorbance values seems to be lower than they expected when compared with CR-alone curve. This may indicate that the experimental protocol for CR assay is sufficiently optimized and requires further improvement.
One of the critical points could be adequate control of sample pH as CR absorbance depends on pH. Another point is usage of NanoDrop spectrophotometer to conduct CR assay. This device equipped with 1mm optical path unit may not be suitable for this analysis. Under this condition, OD of 20 uM CR solution will not exceed 0.074. This is very low value for this assay and thus the measurements may have significant error.
2. CR has an absorbance maximum at 498 nm (pH 7.4, phosphate buffer saline). Nevertheless, according to Figure 2B, it is approximately 470 nm. Probably, axis X was scaled in a wrong way.
One more issue regarding Materials and Methods section has to be pointed out. I apologize for overlooking this earlier.
3 Congo red assay protocol description (page 15, lines 578-587) needs to be clarified. Usually, Congo red measurements are taken at pH 7.4. RHIM-containing proteins were dialysed against the buffer pH 7.4, whereas insulin monomers and fibrils were prepared in buffer pH 2.0. Please, give an appropriate reference or describe in more details how CR solution was prepared (what buffer did you use to prepare the stock, did you filtrate it or not, did you check the real concentration of CR in the stock) as well as how CR-insulin complexes were prepared for the measurement in terms of buffer pH.
Conclusion:
Extensive revision:
1. Birefrigence analysis of Ub-TRIF proteins;
2. Optimization of CR test
3. Further correction of Figure 2B if necessary
4. Revision of CR assay description in Materials and Methods section.
Author Response
Response regarding the effect of AAAA mutation on TRIF RHIM
Respectfully, we have sought to convey our results showing that the AAAA mutation to the RHIM abolishes the ability of the protein to form the long, regular fibril morphology characteristic of amyloid. However, even though we show that the mutant only assembles into small irregular aggregates, nonetheless we find that these do give rise to ThT fluorescence and Congo red signals in solution that are similar to those generated by other cross-beta-containing structures and the wild type protein. Throughout the manuscript text we have not referred to the assemblies formed by the mutant protein as amyloid but instead, have used the description “contains some elements of cross-beta architecture”. We have amended the text to clarify this point.
We have studied this AAAA mutation because of the functional data in the literature reporting on the lack of necroptosis signaling when this mutation is introduced. Our data are consistent with this mutation abolishing both a fibrillar morphology AND interaction with other RHIM proteins with functional consequences.
1. Congo red assay results
We appreciate the points raised by this reviewer regarding the CR results but we do observe variation in the absorbance and have represented this faithfully. Large aggregates sediment rapidly in the cuvette and even with mixing immediately before measurement, this is one source of variation in the amount of absorbing material in the beam path.
The measurements were conducted in a Nanodrop spectrophotometer 2000C with built-in cuvette facility and the measurements were made using cuvettes with a pathlength of 1 cm. We did not use the pedestal function that has 1mm optical path.
2. We apologise for this error in preparation of the figure and appreciate the reviewer drawing our attention to this. Panel B has been correctly represented in the new version and the absorbance maximum for CR is close to 500 nm as described.
3. Yes, the Congo red stock solution was dissolved in the buffer and filtered. Additional detail has now been added to the Methods section. The insulin fibril and monomer samples were prepared at the same pH to allow appropriate comparison.
Reviewer 2 Report
I thank the authors for addressing my concerns.
I am still worried about the way the protein concentration was determined in this paper. The BCA Kit the authors used has only a tolerance for urea concentration up to 3 M urea. I would think that this will introduce systematic error, which should not effect the conclusion of this work, but could lead to problems when reproducing the work.
I would like the authors to put more thoughts into how protein concentration is determined in their further scientific work. For example Gill and von Hippel have a good method to determine protein concentration in 6M guanidinium hydrochloride that could have been used.
I appreciate the addition of the experiment in figure S2.
I think the work here is significantly improved from the first version.
Author Response
Thank you for this suggestion. We will look to incorporate this approach in future studies but reiterate that for all protein determination, the protein-containing samples were diluted to yield [urea] <0.1 M before they were tested in the BCA Kit. This is within the range of urea compatible with use of this kit. Additional text has been added to the Methods section to clarify this point.